# EFFICIENT WRAPPER FEATURE SELECTION USING AUTOENCODER AND MODEL BASED ELIMINATION

## ABSTRACT

We propose a computationally efficient wrapper feature selection method - called Autoencoder and Model Based Elimination of features using Relevance and Redundancy scores (AMBER) - that uses a single ranker model along with autoencoders to perform greedy backward elimination of features. The ranker model is used to prioritize the removal of features that are not critical to the classification task, while the autoencoders are used to prioritize the elimination of correlated features. We demonstrate the superior feature selection ability of AMBER on 4 well known datasets corresponding to different domain applications via comparing the accuracies with other computationally efficient state-of-the-art feature selection techniques. Interestingly, we find that the ranker model that is used for feature selection does not necessarily have to be the same as the final classifier that is trained on the selected features. Finally, we hypothesize that overfitting the ranker model on the training set facilitates the selection of more salient features.

## 1 INTRODUCTION

Feature selection is a preprocessing technique that ranks the significance of features to eliminate features that are insignificant to the task at hand. As examined by Yu and Liu (2003), it is a powerful tool to alleviate the curse of dimensionality, reduce training time and increase the accuracy of learning algorithms, as well as to improve data comprehensibility. For classification problems, Weston et al. (2001) divide feature selection problems into two types: $(a)$ given a fixed $k \ll d$, where $d$ is the total number of features, find the $k$ features that lead to the least classification error and $(b)$ given a maximum expected classification error, find the smallest possible $k$. In this paper, we will be focusing on problems of type $(a)$. Weston et al. (2001) formalize this type of feature selection problems as follows. Given a set of functions $y = f(x, \alpha)$, find a mapping of data $x \mapsto (x * \sigma)$, $\sigma \in \{0, 1\}^d$, along with the parameters $\alpha$ for the function $f$ that lead to the minimization of

$$\tau(\sigma, \alpha) = \int V(y, f((x * \sigma), \alpha)) dP(x, y), \tag{1}$$

subject to $\|\sigma\|_0 = k$, where the distribution $P(x, y)$ - that determines how samples are generated - is unknown, and can be inferred only from the training set, $x * \sigma = (x_1 \sigma_1, \ldots, x_d \sigma_d)$ is an elementwise product, $V(\cdot, \cdot)$ is a loss function and $\| \cdot \|_0$ is the $L_0$-norm.

Feature selection algorithms are of 3 types: Filter, Wrapper, and Embedded methods. Filters rely on intrinsic characteristics of data to measure feature importance while wrappers iteratively measure the learning performance of a classifier to rank feature importance. Li et al. (2017) assert that although filters are more computationally efficient than wrappers, due to the absence of a learning algorithm that supervises the selection of features, the features selected by filters are not as good as those selected by wrappers. Embedded methods use the structure of learning algorithms to embed feature selection into the underlying model to reconcile the efficiency advantage of filters with the learning algorithm interaction advantage of wrappers. As examined by Saeys et al. (2007), embedded methods are model dependent because they perform feature selection during the training of the learning algorithm. This serves as a motivation for the use of wrapper methods that are not model dependent. Weston et al. (2001) define wrapper methods as an exploration of the feature space, where the saliency of subsets of features are ranked using the estimated accuracy of a learning algorithm. Hence, $\tau(\sigma, \alpha)$ in (1) can be approximated by minimizing

$$\tau_{wrap}(\sigma, \alpha) = \min_{\sigma} \tau_{alg}(\sigma), \tag{2}$$

subject to $\sigma \in \{0,1\}^d$, where $\tau_{alg}$ is a classifier having estimates of $\alpha$. Wrappers methods can further be divided into three types: Exhaustive Search Wrappers, Random Search Wrappers, and Heuristic Search Wrappers. We will focus on Heuristic Search Wrappers that iteratively select or eliminate one feature at each iteration because unlike Exhaustive Search Wrappers, they are more computationally efficient and unlike Random Search Wrappers, they have deterministic guarantees on the set of selected salient features, as illustrated in Hira and Gillies (2015).

## 1.1 MOTIVATION

### 1.1.1 RELEVANCE AND REDUNDANCY

We hypothesize that the saliency of features is determined by two factors: Relevance and Redundancy. Irrelevant features are insignificant because their direct removal does not result in a drop in classification accuracy, while redundant features are insignificant because they are linearly or non-linearly dependent on other features and can be inferred - or approximated - from them as long as these other features are not removed. As shown in Fig. 1 and detailed by Guyon et al. (2008), one does not necessarily imply the other. Fig. 1 (a) shows 2 highly redundant features (represented by $x$ and $y$ values) that are both relevant, as removal of any of the features will lead to an inability to classify the data and (b) shows 2 features, where the removal of any one feature does not deter the classification ability, and thus any of them is irrelevant given the other. However, they are not highly redundant as the value of any of them cannot be well approximated using the other.

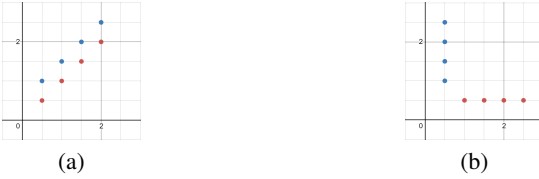

|       |       |
|:-----:|:-----:|
| (a)   | (b)   |

Figure 1: (a): Redundant but not irrelevant; (b): Irrelevant but not redundant

Filter methods are better at identifying redundant features while wrapper methods are better at identifying irrelevant features, and this highlights the power of embedded methods as they utilize aspects of both in feature selection as mentioned in Bolón-Canedo et al. (2013). Since most wrapper methods do not take advantage of filter method based identification of redundant features, there is a need to incorporate a filter based technique to identify redundant features into wrapper methods, which we address using autoencoders.

### 1.1.2 TRAINING THE CLASSIFIER ONLY ONCE

Although wrapper methods often deliver higher classification accuracies compared to filter methods, their computational complexity is often significantly higher because the classifier needs to be trained for every considered feature set at every iteration. For greedy backward elimination wrappers, the removal of one out of $d$ features requires removing each feature separately and training the classifier with the remaining $d - 1$ features and testing its performance on the validation set. The feature whose removal results in the highest classification accuracy is removed because its removal caused the least impact on performance. This is the procedure followed by most backward feature selection algorithms such as the Recursive Feature Elimination (RFE) method proposed by Guyon et al. (2002). For iterative greedy elimination of $k$ features from a set of $d$ features, the classifier has to be trained for $\sum_{i=1}^{k}(d-i+1)$ times, which poses a practical limitation when the number of features is large. Also, the saliency of the features selected is governed by how good the classifier that ranks the features is, and as such, we need to use state-of-the-art classifiers for ranking the features (CNNs for image data, etc.). These models are often complex and thus, consume a lot of training time which implies a trade-off between speed and the saliency of selected features. We address this issue by training the the feature ranker model only once.

## 2 STATE OF THE ART

In this section, we describe top-notch fast/efficient feature selection methods that we will be comparing our proposed method to. With the exception of FQI, the implementations of these methods can be found in the scikit-feature package created by Li et al. (2017) .

### 2.1 FISHER SCORE

The Fisher Score encourages selection of features where feature values within the same class are similar and feature values belonging to different classes are distinct. Duda et al. (2012) define the Fisher Score for feature $f_i$ as

$$FisherScore(f_i) = \frac{\sum_{j=1}^{c} n_j(\mu_{ij} - \mu_i)^2}{\sum_{j=1}^{c} n_j\sigma_{ij}^2}, \tag{3}$$

where $c$ is the number of classes, $n_j$ represents the number of training examples in class $j$, $\mu_i$ represents the mean value of feature $f_i$, $\mu_{ij}$ represents the mean value of feature $f_i$ for training examples in class $j$, and $\sigma_{ij}^2$ represents the variance of feature $f_i$ for training examples in class $j$.

### 2.2 CMIM

Conditional Mutual Information Maximization (CMIM) is a fast feature selection method proposed in Vidal-Naquet and Ullman (2003) and Fleuret (2004) that iteratively selects features while maximizing the Shannon mutual information function between the feature being selected and class labels, given already selected features. Li et al. (2017) define the CMIM score for feature $f_i$ as

$$J_{\text{CMIM}}(f_i) = \min_{X_j \in \mathcal{S}} [I(X_i; Y | X_j)], \tag{4}$$

where $\mathcal{S}$ is the set of currently selected features, $X_i$ is the random variable representing the value of feature $f_i$, and $I(X; Y | Z)$ is the conditional mutual information between discrete random variables $X$ and $Y$ given a random variable $Z$. Also, we use empirical distributions to compute mutual information functions based on the training set.

### 2.3 RFS

Efficient and Robust Feature Selection (RFS) is an efficient feature selection method proposed by Nie et al. (2010) that exploits the noise robustness property of the joint $\ell_{2,1}$-norm loss function, by applying the $\ell_{2,1}$-norm minimization on both the loss function and its associated regularization function. Li et al. (2017) define RFS's objective function as

$$\min_{W} \|XW - Y\|_{2,1} + \alpha \|W\|_{2,1}, \tag{5}$$

where $X$ is the data matrix, $Y$ is the one-hot label indicator matrix, $W$ is a matrix indicating feature contributions to classes, and $\alpha$ is the regularization parameter. Features are then ranked by the $\ell_2$ norm values of the corresponding row in the optimal matrix $W$. The value of $\alpha$ for our experiments was chosen by performing RFS on a wide range of values and picking the value that led to the highest accuracy on the validation set.

### 2.4 FQI

Feature Quality Index (FQI) is a feature selection method proposed by De Rajat et al. (1997) that utilizes the output sensitivity of a learning model to changes in the input, to rank features. FQI serves as the main inspiration for our proposed method and as elaborated in Verikas and Bacauskiene (2002), the FQI of feature $f_i$ is computed as

$$FQI(f_i) = \sum_{j=1}^{n} \|o_j - o_j^i\|^2, \tag{6}$$

where $n$ is the total number of training examples, $o_j$ is the output of the model when the $j^{th}$ training example is the input, and $o_j^i$ is the output of the neural network when the $j^{th}$ training example, with the value of the $i^{th}$ feature set to 0, is the input.

## 3 AMBER

### 3.1 SENSITIVITIES OF WEIGHTS TO FEATURES

During backpropagation, higher losses in the output layer tend to manifest as a result of larger changes, from the optimal, in the values of the weights in the neural network. Generally, the magnitudes of weights connected to the neurons in the input layer that correspond to more salient features tend to have larger magnitudes and this has been extensively documented by Bauer Jr et al. (2000), Belue and Bauer Jr (1995), and Priddy et al. (1993). Similar to FQI, we measure the relevance of each feature by setting the input to the neuron corresponding to that feature to $0$. This essentially means that the input neuron is dead because all the weights/synapses from that neuron to the next layer will not have an impact on the output of the neural network which implies that the model that has been trained with all features will experience a degradation in its ability to classify the input data. Since more salient features possess weights of higher magnitude, these weights influence the output to a greater extent and setting the values of more salient features to $0$ in the input will result in a greater degradation in the ability of the neural network to classify the input compared to when the same is done for less salient features. This can be measured by the loss given in the output layer where a greater loss corresponds to a greater degradation. This is the basis of the Weight Based Analysis feature selection methods outlined by Lal et al. (2006). We further note that we normalize the training set before training by setting the mean of each feature to $0$ and the variance to $1$, so that our *simulation of feature removal* is effectively setting the feature to its mean value for all training examples. To summarize, the pre-trained neural network ranker model prioritizes the removal of features that are non-relevant to the classification task by simulating the removal of a feature and computing the resulting loss of the model. Features whose removal results in a lower loss are less relevant and we will refer to the loss value of this model as a feature's **Relevance Score**.

### 3.2 AUTOENCODERS REVEAL NON-LINEAR CORRELATIONS

It is possible that the weights connected to less salient features possess high magnitudes. This can take place as these features are redundant in presence of other salient features as described in Sec. 1.1.1. Hence, we use a filter based technique that is independent of a learning algorithm to detect these redundant features. We experimented with methods like PCA as detailed by Witten et al. (2009) and correlation coefficients as detailed by Mitra et al. (2002) but these methods revealed only linear correlations in data, which is why we introduced autoencoders into the proposed method because they reveal non-linear correlations as examined by Han et al. (2018), Balın et al. (2019), and Sakurada and Yairi (2014). To eliminate one feature from a set of $k$ features, we train the autoencoder with one hidden layer consisting of $k-1$ hidden neurons as illustrated in Fig. 2 using the normalized training set.

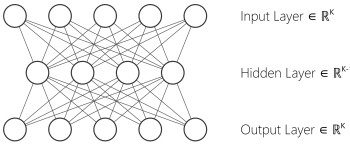

Figure 2: Autoencoder trained to eliminate 1 feature from a set of $k$ features

This hidden layer can either be dense, LSTM, or of other types depending on the data we are dealing with. To evaluate a feature, we set its corresponding values in the training set to $0$ and pass the set into the autoencoder. We then take the Mean Squared Error (MSE) between the output and the original input before the values corresponding to the evaluated feature were set to $0$, and perform this for each of the $k$ features separately. The feature with the lowest MSE is the least salient feature because the other features in the latent space consisting of $k-1$ neurons were able to compensate - with least reconstruction error - for the loss of this feature. We refer to this MSE as a feature's **Redundancy Score**.

### 3.3 Using Transfer Learning to prevent retraining

To eliminate $k$ out of $d$ features, we pick a state-of-the-art neural network model for the dataset and train it on the training set using part of it as the validation set. We call this model the **Ranker Model** (RM) as it allows us to rank the saliency of the features. Next, we set the input for each of the $d$ features in all the examples of the training set to $0$ one at a time in a round-robin fashion to obtain a list of $d$ Relevance Scores after evaluating the modified training sets on the RM. Additionally, we train the autoencoder with one hidden layer consisting of $d - 1$ hidden neurons and pass the same modified training sets through the autoencoder to obtain $d$ Redundancy Score for each of the $d$ features. We then divide the Relevance and Redundancy Scores by their corresponding ranges so that they both contribute equally to the final decision and add the corresponding Relevance and Redundancy Scores to obtain the **Saliency Score**. The feature with the lowest Saliency Score is then eliminated from the training set. In the context of the RM, elimination means that that feature is permanently set to $0$ for all the examples in the training set. Thus, we can reuse the same RM on the next iteration of AMBER. In the context of the autoencoder, elimination means that that feature is permanently removed from the training set for all the examples. This entire process is done iteratively $k$ times to eliminate $k$ features. AMBER uses the RM and autoencoders to examine both relevance and redundancy relationships among features in the training data that they are already fit for, to iteratively eliminate features. The pseudocode for AMBER is described in Algorirthm 1. It is important to note that a feature selection algorithm can be implemented either to remove a specified number of features or to stop when the accuracy of the learning algorithm ceases to increase depending on the application. We chose the former method to specifically take into account the scenarios when this accuracy decreases before it increases later on in the feature selection process. Further, it is not difficult to implement a variant of AMBER according to the latter objective.

---

**Algorithm 1:** AMBER Algorithm for Feature Selection

---

**Inputs**: k: Number of features to be eliminated; trainSet: Training Dataset;
**Outputs**: featList: List of $k$ eliminated features

**function** AMBER(k, trainSet)
Train state of the art RM using trainSet
Initialize featList to empty list
**for** $i = 1$ *to* $k$ **do**

    Set dmTrainSet as trainSet where all features in featList are set to 0
    Set autoTrainSet as trainSet where all features in featList are removed
    Train autoencoder with one hidden layer containing $d - i$ units using autoTrainSet
    **for** $j = 1$ *to* $d - i + 1$ **do**

        Record loss of RM when dmTrainSet is evaluated after setting feature $j$ to 0
        Set cTrainSet as autoTrainset where feature $j$ is set to 0
        Record MSE of autoTrainSet and output of autoencoder when cTrainSet is evaluated

    Normalize RM losses and MSEs and add corresponding values
    Sort and add lowest scoring feature to featList
**return** (featList)

---

Once the final set of $k$ features to be eliminated are determined, they are completely removed from both the training and testing sets. The final architecture is then trained on the training set consisting of $d - k$ features and tested on the testing set also consisting of $d - k$ features.

## 4 Results

### 4.1 Experimental Setup

We used a GPU server equipped with 3 Nvidia Tesla P100 GPUs, each with 16 GB of memory and used Keras with a TensorFlow backend as the environment of choice. With the exception of the RadioML2016.10b dataset for which we used all 3 GPUs, we only used 1 GPU for training. The experiments were performed 3 times and the average accuracies were plotted at each feature count in Fig. 3. The source code for AMBER, links to the datasets considered, and the error bars for the comparison plots are available at `https://github.com/amber-iclr/AMBER`.

## 4.2 DATASETS

Each dataset corresponds to a different domain of data and encompasses both low and high dimensional data to demonstrate the versatility of AMBER. The final models that are trained on the set of selected features are common across all the feature selection methods that are compared and are trained until early stopping is achieved with a patience value of 5 to ensure that the comparisons are fair. For all the datasets, the softmax activation function is applied to the output layer with the cross-entropy loss function. The test split used for the Reuters and the Wisconsin Breast Cancer dataset is $0.2$ while the test split used for the RadioML2016.10b dataset is $0.5$. The plots in Fig. 3 were jagged when feature counts in decrements of 1 were plotted and thus, in the interest of readability, we plotted them in larger feature count decrements as specified for each dataset. Finally, to demonstrate that the final model does not necessarily have to be the same as the RM used by AMBER, we used different models as the final model and the RM for the MNIST and RadioML2016.10b datasets.

**MNIST** is an image dataset created by LeCun et al. (1998) consisting of 60000 28x28 grayscale images with 10 classes, each belonging to one of the 10 digits, along with a test set that contains 10000 images of the same dimensions. The total number of features here is 784. The Ranker Model here is a CNN consisting of 2 convolutional layers, a max pooling layer, and 2 dense layers, in that order. The convolutional layers have 32 and 64 filters, in order of depth, with kernel sizes of (3x3) for both layers. The max pooling layer has a of pool size of (2x2) and the dense layers have 128 and 10 (output layer) neurons, with Softmax activation for the output layer. ReLU is applied to the remaining layers. The final model used is an MLP model consisting of 3 fully connected layers with 512, 512, and 10 (output layer) neurons. ReLU is applied to each of the layers with 512 neurons and these layers are followed by dropout layers with a dropout rate of $0.2$.

**Reuters** is a text dataset from the Keras built-in datasets that consists of 11228 newswires from Reuters with 46 classes, each representing a different topic. Each wire is encoded as a sequence of word indices, where the index corresponds to a word's frequency in the dataset. For our demonstration, the 1000 most frequent words will be used and thus, the total number of features is 1000. The Ranker Model and the final model are the same MLP model consisting of 2 fully connected layers with 512 and 46 (output layer) neurons with ReLU and Softmax activation functions, respectively.

**Wisconsin Breast Cancer** is a biological dataset created by Street et al. (1993) that consists of features that represent characteristics of cell nuclei that have been measured from an image of Fine Needle Aspirates (FNAs) of breast mass. The dataset consists of 569 examples that belong to 2 classes: malignant or benign. The total number of features here is 30. The Ranker Model and the final model are the same MLP model consisting of 4 fully connected layers with 16, 8, 6, and 1 (output layer) neurons, in order of depth. Softmax is applied to the output layer while ReLU is applied to the remaining layers.

**RadioML2016.10b** is a datset of signal samples used by O'Shea et al. (2016) that consists of 1200000 128-sample complex time-domain vectors with 10 classes, representing different modulation types. It consists of 20 Signal to Noise Ratio (SNR) values ranging from -20 dB to 18 dB in increments of 2 dB; we only choose the results of the 18 dB data for better illustration. Each of the 128 samples consists of a real part and a complex part and thus, the input dimensions are 2x128, where the total number of features is 256. This dataset has the unique property that only pairs of features (belonging to the same sample) can be eliminated. AMBER, like FQI, is powerful in such situations as the pairs of features can be set to $0$ to evaluate their collective rank. This is also useful in the case of GANs, where sets of pixels/features in a 2-D pool need to be evaluated to craft adversarial attacks as elaborated by Papernot et al. (2016). The other feature selection methods fail in this case because they account for feature interactions between the pairs of features as well, which is one reason for why AMBER outperforms them as it does not. For the other methods, to eliminate pairs of features belonging to the same sample, we simply added the scores belonging to the two features to obtain a single score for each sample. The Ranker Model used here is a CLDNN while the final model used is a ResNet, both of which are described in Ramjee et al. (2019).

## 4.3 CLASSIFICATION ACCURACIES

The final models' classification accuracy plots using the selected features can be observed in Fig. 3. We observe the impressive performance delivered by AMBER that generally outperforms that of all 4 considered methods, particularly when the number of selected features becomes very low (about

99% average accuracy with 4 out of 30 features for the Cancer dataset and about 95% average accuracy with 16 out of 128 samples for the RadioML dataset). The comparisons of the accuracies of the final models using the top 25% and 10% of features are given in Table 1. The results in the last two rows refer to using a version of AMBER without the Autoencoder's redundancy score, and another version of AMBER where the ranker model is retrained in every iteration, respectively. Note from the depicted results (purple curve in the figures) how solely relying on the RM significantly reduces AMBER's performance, which validates our intuition about the benefit of using the Autoencoder to capture correlations to reduce the generalization error. Further, we observe how negligible gains are achieved when retraining the RM in every iteration, which comes at a significant computational cost, as demonstrated in Table 2, which validates our intuition about simulating the removal of features without retraining for computational efficiency while maintaining good performance.

Table 1: Accuracy Comparisons

| Method | Avg. accuracy with top 25% features (%) | | | | Avg. accuracy with top 10% features (%) | | | |
|---|---|---|---|---|---|---|---|---|
| | MNIST | Reuters | Cancer | RadioML | MNIST | Reuters | Cancer | RadioML |
| Fisher | 96.28 | 68.00 | 96.78 | 82.96 | 88.37 | 51.21 | 92.40 | 73.42 |
| CMIM | 98.06 | 74.14 | 95.61 | 80.74 | 96.38 | 71.04 | 90.64 | 70.22 |
| RFS | 97.17 | 78.94 | 92.40 | 87.44 | 89.46 | 77.11 | 91.81 | 75.85 |
| FQI | 97.90 | 73.59 | 87.13 | 91.70 | 95.49 | 68.20 | 76.32 | 83.57 |
| AMBER | 98.08 | 79.84 | 97.37 | 98.78 | 97.21 | 77.55 | 96.78 | 95.15 |
| - Relevance | 97.38 | 76.60 | 94.04 | 95.21 | 93.29 | 73.45 | 89.65 | 89.54 |
| - Retraining | 98.37 | 81.25 | 98.25 | 99.73 | 97.21 | 78.11 | 97.37 | 97.49 |

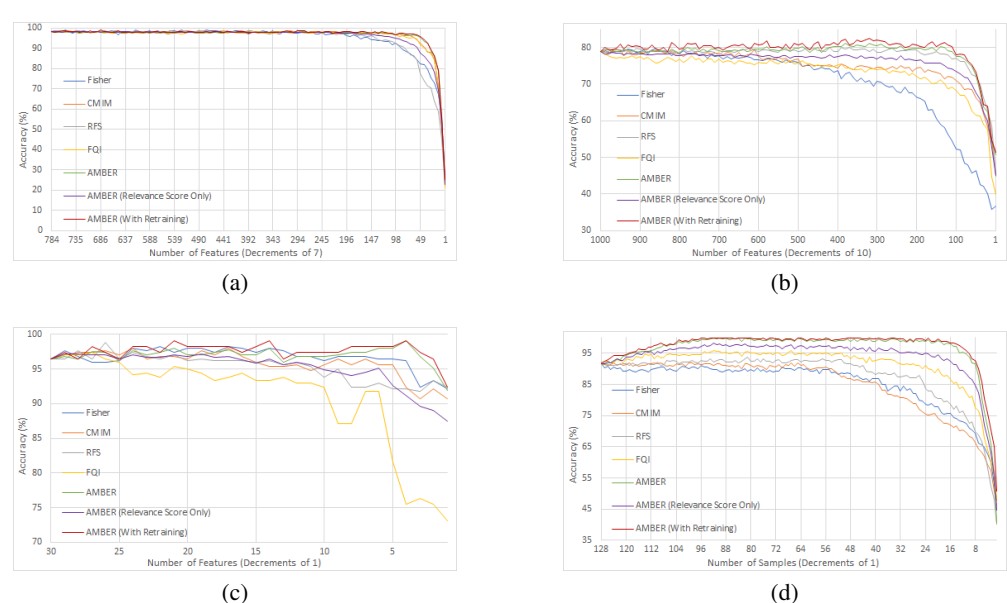

Figure 3: Accuracy vs Feature Count plots for the final models trained with the selected features for the (a) MNIST, (b) Reuters, (c) Wisconsin Breast Cancer, and (d) RadioML2016.10b datasets.

Table 2: Time needed to rank all features in Seconds.

| Method | MNIST | Reuters | Cancer | RadioML |
|---|---|---|---|---|
| AMBER | 10552.24 | 21710.78 | 40.04 | 26417.53 |
| - Retraining | 24202.66 | 29005.08 | 739.01 | 42533.27 |

## 5   DISCUSSION

In Sec. 3 and as evidenced by Wang et al. (2004), we elaborated on how more salient features possess higher magnitudes of weights in the input layer than features that are less salient, which is the property of neural networks that serves as the basis for AMBER. The performance of AMBER heavily depends on the performance of the RM that ranks the features. In some cases, however, even the state-of-the-art models do not have high classification accuracies. In such cases, we can obtain better feature selection results with AMBER by overfitting the RM on the training set.

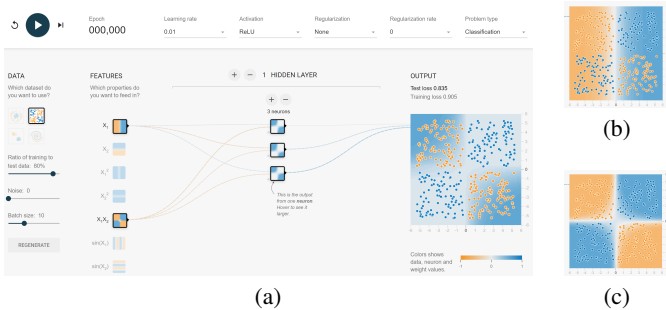

Figure 4: (a): Toy example from TensorFlow Playground; (b) Feature 1 and (c) Feature 2.

We will demonstrate this using the toy example illustrated in Fig. 4 that portrays the architecture used for the RM along with the corresponding hyperparameters. Feature 1 is $x_1$ and feature 2 is $x_1x_2$. Feature 2 is more salient than feature 1 as it is able to form a decision boundary that allows for better classification (see (b) and (c)). Each of these features have 3 weights in the input layer and as expected, the weights connected to feature 2 manifest into weights of higher average magnitude than those belonging to feature 1 as shown in Fig. 5. As the number of training epochs increases, the difference in the average magnitudes of the weights increases. This implies that the RM will be able to better rank the saliency of features as the difference between the Relevance Scores of more and less salient features increases. Thus, we can overfit the RM on the training set by training it for a large number of epochs without regularization to enable better feature selection.

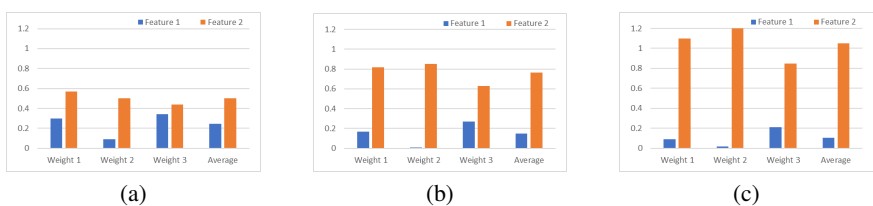

Figure 5: Input layer weight magnitudes after training for (a) 10, (b) 100, and (c) 1000 epochs.

## 6   CONCLUDING REMARKS

AMBER presents a valuable balance in the trade-off between computational efficiency in feature selection, in which filter-based methods excel at, and performance (i.e. classification accuracy), in which traditional wrapper methods excel at. It is inspired by FQI with two major differences: 1- Instead of making the final selection of the desired feature set based on simulating the model's performance with elimination of only a single feature, the final model's performance in AMBER is simulated with candidate combinations of selected features, 2- The autoencoder is used to capture redundant features; a property that is missing in FQI as well as most wrapper feature selection methods. However, we found AMBER to require slightly larger computational time than the considered 4 state-of-the-art methods, and we also found it to require far less time than state-of-the-art wrapper feature selection methods, as it does not require retraining the RM in each iteration.

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
