# OpenReview forum: "Efficient Wrapper Feature Selection using Autoencoder and Model Based Elimination"
_ICLR.cc/2020/Conference — Reject_

### Official Review · AnonReviewer1 · 2019-10-22
**Official Blind Review #1**

**Rating:** 1

**Review:**

In this paper, the authors present an iterative approach for feature selection which selects features based both on the relevance and redundancy of each feature. The relevance of each feature is determined using a mild variant of the Feature Quality Index; essentially, the relevance is computed as the loss in model performance when setting each feature value to the mean and measuring the change in performance. Similarly, the redundancy of each feature is determined by comparing the reconstruction loss of an autoencoder when setting the feature value to its mean for all training samples. These two values are combined to give a single score for each feature at each iteration. The feature with the worst value is removed. A limited set of experiments suggests the proposed approach mildly outperforms other efficient feature selection methods.

Major comments

The paper does not include relevant, recent work on using autoencoders for feature selection, such as [Han et al., ICASSP 2018; Balın et al., ICML 2019], among others. Thus, it is difficult to discern how this paper either theoretically or empirically advances the state of the art.

I found the proposed approach to efficient feature selection reasonable. However, there is no theoretical justification for the approach. Thus, I would expect a thorough empirical analysis. Only a few limited experiments on toy datasets (and one slightly more challenging one) are given.

The paper is not well-written. For example, it seems as though the proposed approach is not applicable to datasets with categorical features. It is not obvious (and, presumably, would need to be shown empirically) if the mode could be used to replace categorical values analogously to how the mean is used for real-valued features. Alternatively, one could imagine one-hot encoding the categorical variables and grouping them in some manner similar to that used for the RadioML pairs (since the one-hot values are obviously highly correlated). However, the authors do not address these issues.

Similarly, the entire discussion in Section 3 seems to assume the ranker model will be some sort of neural network. However, as far as I can tell, the ranker model is treated as a black box, so it could easily be some random forest model, etc. If there are some implicit assumptions that the ranker model is a neural network, this should be made explicit; if not, the discussion should be revised (and, of course, non-neural models should be used in the experiments).

The approach seems to heavily depend on the ability of the autoencoder to reconstruct the input; however, it is unclear how the structure/capacity of the autoencoder affects the performance of the algorithm. For example, the authors propose a relatively simple structure, presumably to maintain computational efficiency. It would be interesting to explore more deeply how autoencoders with more capacity impact the results.

It is unclear why the autoencoder is retrained at each step compared to just setting the removed feature values to the respective means, as is done with the ranker model.

Clearly, the relevance and redundancy scores could be weighted unequally when selecting the feature to remove. It would be interesting to explore how different combinations affect the results.

It seems that the experiments only consider backward feature selection approaches. Including forward feature selection approaches would add useful context for how the proposed approach compares to other strategies.

Minor comments

The cross-validation scheme used is not clear. While the authors mention that three runs are used to estimate performance variance, they do not describe if this is 3-fold cross validation, some Monte Carlo cross validation, or if the same splits are used all three times and just the random seeds are different.

While methods like RFE have significantly higher computational cost than the methods considered here, it would be helpful to include it for at least one of the datasets to provide context on how much the less costly methods “lose”.

What is the overlap in the selected features? both among the different methods and among the different folds for the same method.

How were the hyperparameters for the various models chosen?

Typos, etc.

The references are not consistently formatted.

The Section 2 headers all have an unnecessary “0” in them (e.g., “2.0.1”).

Table 1 should include the standard deviations.


**Experience Assessment:**

I have published one or two papers in this area.

**Review Assessment: Checking Correctness Of Derivations And Theory:**

I assessed the sensibility of the derivations and theory.

**Review Assessment: Checking Correctness Of Experiments:**

I carefully checked the experiments.

**Review Assessment: Thoroughness In Paper Reading:**

I read the paper at least twice and used my best judgement in assessing the paper.

---

> ### Author Response · Authors · 2019-11-15
> **Response to Official Blind Review #1**
>
> - Thank you for the review! We specially thank you for bringing our attention to important recent work on using Autoencoders for unsupervised feature selection. We have cited the mentioned work with brief description – due to the page limit. We would like to note that while the goal of these works was unsupervised feature selection, here we have a classification task and our goal is to select the important features with respect to that task. It is possible, however, to replace our autoencoder with more complex designs that are detailed in these works, but since this was our first attempt at an AMBER-like framework, we chose the simplest architecture that delivers good results, as a proof of concept. We agree that these could be exciting venues of research for extending our and these works.
>
> - Theoretical justification is out of scope of this work. We used 4 standard datasets belonging to 4 different applications to demonstrate the generality of the proposed algorithm.
>
> - We believe that the Reuters dataset – one of the 4 used for testing AMBER – is based on categorical features. This may have been a mistake in the review.
>
> - It is true that the focus of this work is on neural network ranker models. This is because we exploit the transferability property to lesser dimensions by “shutting down” features. We further clarified that in the revised submission.
>
> - We agree with your comment: “It would be interesting to explore more deeply how autoencoders with more capacity impact the results”. As illustrated in the first point of this response, since this is a first attempt at the proposed framework, we chose the simplest architecture that works as a proof of concept.
>
> - The autoencoders are retrained in each step, because we use a different autoencoder architecture (different number of input and hidden layer units) for each step. This is unlike the fixed-architecture ranker model used for classification. It does not look straightforward to us how to use one autoencoder architecture for all steps (for example, what would be the number of units in the bottleneck layer?).
>
> - In our trials, we could not find an advantage to unequal relevance and redundancy score weights, but we agree that this is an interesting topic of research.
>
> - We believe that our method is suitable for backward feature selection, as the interactions between important features are maintained in every step, and hence always contribute to the ranking of features. On the other hand, with greedy forward feature selection, these interactions are no longer fully captured: Consider a simple forward feature selection example where we have already selected the most important feature a, and simulate the second selection of candidate features b and c; that simulation does not take into how selecting b would affect the relevance of c, and how selecting c would affect the relevance of b. This could be problematic since at this early stage, all these features could be very important to the task, and their combinations should be fully considered.
>
> - We used K-fold cross validation, but we found it to lead to negligible results compared with fixing a validation set, so for simplicity of presentation, we replaced the term “cross-validation set” with “validation set” in the revised submission. Thanks for the careful observation!
>
> - A method like RFE could take excessive amount of computation times to select features, particularly with a large number of features, and when it’s desired to select a small set (using backward elimination), as it requires retraining the model in every iteration. To capture the computational cost of retraining, we report in Table 2 of the revised submission the computation times needed to rank all the features for the different datasets with AMBER, and another version of AMBER that retrains the model in every iteration.
>
> - Thanks for raising the point about the overlap of the selected feature sets. We are working on including these ratios – and investigating potential insights – in the final version.
>
> - The hyperparameters used for training the models, and not mentioned in the main document, are now stated in a text file in the Github folder. Thanks for raising this important point!
>
> - We fixed the references consistency issue in the revised submission.
>
> - We fixed the Section 2 headers issue in the revised submission.
>
> - For standard deviations, we performed the experiments 3 times, and provide the error bars and accuracy results for all 3 experiments in the Github folder. Because of the page limit, it was difficult to include them in the main document.

---

> > ### Comment · AnonReviewer1 · 2019-11-15
> > **Response addresses a few issues, but others remain**
> >
> > I have read the other reviews and authors' responses. The response does clarify several important details about the experimental design, but my basic concerns still remain that there are many empirical choices which need further exploration to justify the described approach. Other reviewers also raised this issue, and the authors acknowledge that theoretical analysis is not in the scope of this paper.
> >
> > So, the additional information about the experiments improve my view of the paper, I still do not believe it is ready for acceptance at this time.

---

### Official Review · AnonReviewer2 · 2019-10-30
**Official Blind Review #2**

**Rating:** 3

**Review:**

The article "Efficient Wrapper Feature Selection using Autoencoder and Model Based Elimination" considers the problem of feature selection for a broad class of machine learning models. The authors argue that it is important to consider the relevance of features for the considered supervised ML problem and redundancy of features. They propose the wrapper feature selection method based on this paradigm and report the results of the experimental comparison of the method with some approaches from the literature.

The proposed AMBER approach consists of 2 parts:
1. The ranking of features based on the sensitivity of some supervised ML model with respect to the particular feature.
2. The ranking of features with respect to their individual impact on the accuracy of the autoencoder trained on the features of the training data set.

The scores obtained on these 2 steps are added and the algorithm iteratively removes features with the lowest total score.

I should note that the proposed approach is very general, but the paper gives very few details on the actual implementation. For example, it seems important to properly normalize relevance and redundancy scores before computing the final score but the paper doesn't discuss this issue. Also, there are many possible ways to compute losses. For example, one can use training or validation sets for that but the authors choose the training set without motivation.

Most importantly, the experimental part of the paper considers just 4 datasets. I believe that the algorithms of such generality should be evaluated on a much broader selection of problems. The most important thing is whether it is possible to select hyperparameters of the method in a way that few human interventions are needed to achieve high-quality results.

Overall, I am very concerned with making the particular instance of the proposed approach working on a vast selection of applied problems. The provided repository with code confirms my concerns as it doesn't provide the single algorithm but rather the collection of scripts tailored for particular problems considered.

To sum up, I think that while the motivation behind the paper is very natural, I am not convinced with experimental results and the overall applicability of the approach.

**Experience Assessment:**

I have read many papers in this area.

**Review Assessment: Checking Correctness Of Derivations And Theory:**

N/A

**Review Assessment: Checking Correctness Of Experiments:**

I assessed the sensibility of the experiments.

**Review Assessment: Thoroughness In Paper Reading:**

N/A

---

> ### Author Response · Authors · 2019-11-14
> **Response to Official Blind Review #2**
>
> Thank you for the review! Please find below responses to each of the points raised:
>
> - We believe that the implementation details were sufficiently described. As illustrated in the draft, the relevance and redundancy scores are normalized by their ranges. Further, the losses are computed based on the training rather than validation dataset for two reasons: a) It is typically larger, allowing for more accurate feature selection, as we found in our experiments. b) As illustrated in the discussion section, overfitting of the ranker model is not necessarily a bad phenomenon for our feature selection method, and that advantage can be best exploited by measuring losses based on the training dataset.
>
> - We believe that the 4 datasets used demonstrate the generality of the algorithm, since they are well known standards and belong to 4 different applications. We are working on including results for the CIFAR-10 dataset as well. We would like to also note that AMBER can be easily modularized to avoid custom-tailoring for each application. Basically, all what is needed to “tailor” it for a new application is to provide the ranker model.

---

### Official Review · AnonReviewer4 · 2019-11-04
**Official Blind Review #4**

**Rating:** 3

**Review:**

This paper proposes a wrapper feature selection method AMBER to use a single ranker model along with autoencoders to perform greedy backward elimination of features. Experimental results on various datasets show that their criterion outperforms other baseline methods. Generally, the paper is well written and easy to follow. However, the idea is simple and the originality seems incremental.

First, although taking advantage of the power of neural network to help select better features sounds interesting, it is important for the author to discuss the benefit of doing it. As mentioned by the author, neural network essentially has the ability of selecting features. Instead of selecting features with AMBER explicitly, a more straightforward way is using all features as input and solving the downstream task with deep learning model. Feature selection will be automatically conducted during the learning process. It will be better if the author can explain more about the benefit and motivation of introducing feature selection explicitly in this scenario.

Second, the author proposed to use an autoencoder with one hidden layer consisting of d−1 hidden neurons to calculate feature’s redundancy score. Is there any specific reason for including d-1 hidden neurons. It will be better if the author can give some theoretical analysis of it.

Third, the author calculates the redundancy score and relevance score independently and combine them together to obtain the saliency score. However, it seems unreasonable to regard relevance and redundancy as two independent factors and stiffly combine them. For example, a feature can be both relevant and redundant. Should we eliminate it? How the proposed method solve this case?



**Experience Assessment:**

I have published one or two papers in this area.

**Review Assessment: Checking Correctness Of Derivations And Theory:**

N/A

**Review Assessment: Checking Correctness Of Experiments:**

I assessed the sensibility of the experiments.

**Review Assessment: Thoroughness In Paper Reading:**

I read the paper at least twice and used my best judgement in assessing the paper.

---

> ### Author Response · Authors · 2019-11-08
> **Response to Official Blind Review #4**
>
> Hi,
>
> Thanks much for your review! For the comments in your summary, we believe that simplicity is a good thing, if the proposed method is novel and useful. No justification or explanation for the comment about the work being incremental, so we choose not to respond to it.
>
> For the first detailed comment, the main reason for choosing not to embed feature selection in the neural network classifier – besides using the autoencoder and the explainability advantage of explicitly identifying important features - is that the ranker model can be different from the classifier, more specifically as overfitting the ranker model can be advantageous, since it is only used to distinguish between features, and not for final classification. Please check the discussion section for further illustration.
>
> For the second detailed comment, theoretical analysis is beyond the scope of this work, and we are not claiming an optimality conclusion for this architecture, but it is rather used to affirm the usefulness of the autoencoder’s redundancy score in feature selection, and leads to sufficiently good results in our experimental setup. The intuition behind the d-1 dimension for the autoencoder’s bottleneck layer is that we are using it to assess how successful can the representation be reconstructed in the absence of one input feature, so it first tries to find the best d-1 dimensional representation, and then reconstruct from it. Our empirical trials also suggested this option for the architecture over few other alternatives.
>
> For the third detailed comment, it is true that a feature can be both relevant and redundant, and in this case, its relevance score will suggest its saliency but its redundancy score will not. These two suggestions would then be considered, and depending on the final rank of the feature, it will be determined whether to include it. Having separate relevance and redundancy scores was a simple intuitive option that turned out to deliver impressive results, but we are investigating other more intricate scoring strategies as follow-ups to this work.

---

### Public Comment · ~Ian_Connick_Covert1 · 2019-09-28
**Model retraining, and importance of autoencoder**

I have two questions about your method.

1) What is the impact of not retraining your "ranker model" after each set of feature eliminations? It seems like your method would make suboptimal selections, because the ranker model isn't able to adapt to the removed features.

The basic idea of measuring the loss when a feature is imputed by its mean, which you call a "relevance score," is discussed in a highly cited paper from the 90's, see "Neural Network Feature Selector" (Setiono & Liu, 1997). It's like FQI, but it measures the change in loss instead of the change in the output. However, one difference with this work is that they retrain the model after each elimination, which seems like it would work better.

You could easily adapt your method to include model retraining. It shouldn't change the runtime significantly, because your method already requires retraining the autoencoder at every iteration. If you don't adapt your method, you should at least provide a comparison so you can convincingly claim that retraining isn't necessary.

2) Why is the autoencoder aspect of your method necessary? If a feature is redundant, that should already be apparent from the "relevance score," because the model should be able to make accurate predictions without it.

To make a compelling case for the necessity of the autoencoder, you may consider performing an ablation experiment.

---

> ### Author Response · Authors · 2019-10-01
> **Excellent Suggestions**
>
> Hi Ian,
>
> Thanks for your valuable suggestions. We agree that these are important points, and are currently working on incorporating them by adding:
>
> - Accuracy and Runtime comparisons when the ranker model is retrained in every iteration
>
> - Accuracy comparisons when only the relevance score is used

---

> > ### Author Response · Authors · 2019-10-24
> > **Experiments Done**
> >
> > We have conducted the aforementioned experiments, and will update the draft as soon as the rebuttal period begins. For AMBER with only relevance score (no autoencoder), the accuracy seems to be quite lower, which validates our intuition about the benefit of using the Autoencoder to capture correlations to reduce the generalization error. The entries in the new corresponding row in Table I would be (Fisher-25%: 97.38, Reuters-25%: 76.60, Cancer-25%: 94.04, RadioML-25%: 95.21, Fisher-10%: 93.29, Reuters-10%: 73.45, Cancer-10%: 89.65, RadioML-10%: 89.54).
> >
> > With retraining the ranker model in every iteration, only a slight increase in accuracy is observed. The entries in the new corresponding row in Table I would be (Fisher-25%: 98.37, Reuters-25%: 81.25, Cancer-25%: 98.25, RadioML-25%: 99.73, Fisher-10%: 97.21, Reuters-10%: 78.11, Cancer-10%: 97.37, RadioML-10%: 97.49). However, retraining in every iteration incurs significant computation cost. In our experimental setup described in the draft, the training time increases from 10552 to 24202 seconds for MNIST, from 21710 to 29005 seconds for Reuters, from 40 to 739 seconds for Cancer, and from 26417 to 42533 seconds for RadioML , which validates our intuition about simulating the removal of features without retraining for computational efficiency while maintaining good performance.

---

### Decision · Program_Chairs · 2019-12-19

**Decision:**

Reject

**Comment:**

In this paper the authors propose a wrapper feature selection method that selects features based on 1) redundancy, i.e. the sensitivity of the downstream model to feature elimination, and 2) relevance, i.e. how the individual features impact the accuracy of the target task. The authors use a combination of the redundancy and relevance scores to eliminate the features.

While acknowledging that the proposed model is potentially useful, the reviewers raised several important concerns that were viewed by AC as critical issues:
(1) all reviewers agreed that the proposed approach lacks theoretical justification or convincing empirical evaluations in order to show its effectiveness and general applicability -- see R1’s and R2’s requests for evaluation with more datasets/diverse tasks to assess the applicability and generality of the proposed model; see R1’s, R4’s concerns regarding theoretical analysis;
(2) all reviewers expressed concerns regarding the technical issue of combining the redundancy and relevance scores -- see R4’s and R2’s concerns regarding the individual/disjoint calibration of scores; see R1’s suggestion to learn to reweigh the scores;
(3) experimental setup requires improvement both in terms of clarity of presentation and implementation -- see R1’s comment regarding the ranker model, see R4’s concern regarding comparison with a standard deep learning model that does feature learning for a downstream task; both reviewers also suggested to analyse how autoencoders with different capacity could impact the results.
Additionally R1 raised a concern regarding relevant recent works that were overlooked.
The authors have tried to address some of these concerns during rebuttal, but an insufficient empirical evidence still remains a critical issue of this work. To conclude, the reviewers and AC suggest that in its current state the manuscript is not ready for a publication. We hope the reviews are useful for improving and revising the paper.